# Experimental Investigation of Reflectarray Antennas for High-Power Microwave Applications

**DOI:** 10.3390/mi15030399

**Published:** 2024-03-15

**Authors:** Jianing Zhao, Yongzhen Dong, Hao Li, Tianming Li, Wei Liu, Yihong Zhou, Haiyang Wang, Biao Hu, Fang Li, Keqiang Wang, Bin Qiu

**Affiliations:** 1College of Computer Science and Engineering, Guilin University of Technology, Guilin 541006, China; hprl2103zjn@outlook.com (J.Z.); lexi.f.li@outlook.com (F.L.);; 2Guangxi Key Laboratory of Embedded Technology and Intelligent System, Guilin University of Technology, Guilin 541006, China; 3Yangtze Delta Region Institute (Huzhou), University of Electronic Science and Technology of China, Huzhou 313001, Chinayry935@163.com (H.W.); 18671620028@163.com (B.H.);; 4School of Electronic Science and Engineering, University of Electronic Science and Technology of China, Chengdu 610054, China; 5College of Physics and Electronic Information Engineering, Guilin University of Technology, Guilin 541006, China

**Keywords:** reflectarray antenna (RA), high power microwave (HPM) antenna, power handling capacity

## Abstract

The power capacity of reflectarray antennas (RAs) is investigated through full-wave simulations and high-power microwave (HPM) experiments in this paper. In order to illustrate the results in detail, two RA elements are designed. The simulated power handling capacity of two RA elements are 7.17 MW/m^2^ and 2.3 GW/m^2^, respectively. To further study the HPM RA, two RA prototypes operating at 2.8 GHz are constructed with the aperture size of 1 m × 1 m. Simulations and experimental measurements are conducted for the two prototypes. The experimental results demonstrate that, even when subjected to 1 GW of power, the radiation beam of the RA with the second elements can still propagate in the intended direction. This research will establish a basis for advancing the practicality of RAs in HPM applications.

## 1. Introduction

High-power microwave (HPM) technology has been extensively studied and applied in both military and civilian fields [1]. As a terminal radiating system, the microwave antenna plays a crucial role in HPM systems as it determines the effective utilization of HPM energy. As the national defense security demands increase, research and development in HPM antennas will continue to receive widespread attention and support. Currently, there are several typical forms of HPM antennas, including radial line helical array antennas [2,3,4], slotted waveguide array antennas [5,6], and leaky-wave phased array antennas [7]. However, further development of these antennas is currently constrained by various factors, such as the complexity of the feeding network and machining process.

RAs are a combination of reflector antennas and phased array antennas, employing spatial feeding techniques that obviate the requirement for an auxiliary feeding network [8,9,10,11,12]. They offer several advantages, including low cost, high-precision processing, and the ability to customize the modulation of electromagnetic waves. The limited power capacity of RAs has hindered their effective application in the field of HPM, despite being extensively researched. For example, an investigation was carried out on a dual-band RA based on elliptical patches for HPM applications [13]. Full-wave simulation results revealed that the RA has a power capacity of 10.2 MW and 3.9 MW at 6.2 GHz and 9.3 GHz, respectively. In order to promote the power handling capacity of RAs, currently, one commonly used method is to adopt all-metal RA elements [14,15,16,17,18]. In [14], a helical structure is utilized to design an RA for HPM technology, and the power handling capacity could reach 359 MW. An all-metal element composed of a metal base and radiating structure was proposed to enhance the HPM performance of the RA [15]. The simulation results indicate that the power handling capacity of the RA under vacuum conditions exceeds 5 GW/m^2^. The above results indicate that the method can significantly enhance the power handling capacity of RAs. However, the above-mentioned works are often used in circularly polarized applications. Furthermore, the enhanced power capacity is achieved at the cost of planar characteristics and the ease of processing and integration of RAs, which is not conducive to the development of miniaturized and compact HPM systems. An additional method for enhancing the power capacity of RA involves embedding the microstrip patches within a dielectric medium [19]. By adopting this method, the power capacity of the RA element could be increased by two orders of magnitude at the X-band. In [20], a dielectric-embedded RA with a rotation-type phase control is proposed at 10 GHz. The simulation results show that the power capacity of the proposed RA could reach 1 GW/m^2^. Although this method can effectively enhance the power capacity of RAs, high-power experiments have rarely been conducted on this type of RA.

In conclusion, a detailed experiment investigation at the S-band has been carried out in this paper to explore the performance of RAs under HPMs. In order to illustrate the results, two RA elements are designed and analyzed first. On this basis, two prototypes with an aperture size of 1 m × 1 m are designed, fabricated, and measured based on the above-mentioned elements. The experimental results indicate that the power capacity threshold of the RAs can be effectively determined by observing the flashover phenomenon along the array surface, which is helpful for the application of RAs in the field of HPMs.

## 2. Design and Analysis of the HPM RA Elements

The structure of the two RA elements at the S-band are presented in Figure 1. In order to explore the performance of two RA elements in HPMs, the power capacity level of the elements should be evaluated. The power handling capacity of the RA element is calculated by the following equation [13,21]:(1)$$P_{C}=\frac{{\left(E_{b}\right)}^{2}}{{\left(E_{\max}\right)}^{2}}\cdot P_{in}$$
where *P_C_* is the power handling capacity, *E_b_* is the breakdown threshold value, *E*_max_ is the maximum electric field strength of the RA element, and *P_in_* is the input power (the value is usually 1 W in HFSS). For the RA element based on resonance phase modulation, the maximum electric field strength is typically observed at the resonance point. Figure 2 illustrates the surface electric field distributions of two elements at the resonance points. It can be clearly seen that the maximum electric field strengths of two RA elements are 0.28 × 10^5^ V/m and 0.1565 × 10^4^ V/m, respectively. It should be noted that the breakdown threshold in air is 3 MV/m [13,21,22,23,24]. Consequently, applying Equation (1), the theoretical power capacity and power density of RA element I can be calculated as 11.48 KW and 7.17 MW/m^2^. Similarly, the power handling capacity and power density of RA element II are 3.67 MW and 2.3 GW/m^2^, respectively.

## 3. Fabrication and Measurement of the RAs

To verify the actual performance, two of RAs with both elements have been designed with the configuration in Figure 3, the normalized radiation patterns are shown in Figure 4. It can be seen clearly that the two main beam radiation directions are consistent with the desired one. The fabricated prototypes are illustrated in Figure 5. For ease of fabrication, the entire RA surface is constructed by assembling 25 subarrays, each measuring 200 mm × 200 mm. All subarrays are spliced and installed by screws made of PTFE. In order to verify the radiation characteristics and power handling capacity of the two RAs, detailed experimental tests are carried out.

### 3.1. Directivity Testing under Low Power

For the RA, a directivity test is a crucial step to verify the correctness of the design, fabrication, and assembly. Therefore, in the first experiment, an antenna directivity test is conducted under low power conditions. The test block diagram is shown in Figure 6. Due to the limited experimental space, the circle radius radiating from the RA surface center is selected as 151 cm. The feed horn irradiates to the two RAs at 30°. Based on the configuration in Figure 3, it can be observed that the electromagnetic waves emitted by the feed source will propagate outward at a perpendicular angle to the RA surface. This direction corresponds to the position labeled as #0 in Figure 6, commonly referred to as the normal direction. The experimental setups of the two RAs are shown in Figure 7. Table 1 presents the results of the directivity test at a frequency of 2.8 GHz for the two RAs. The table reveals that both RAs demonstrate the maximum radiated power when positioned in the #0 direction. This observation further substantiates the dependability of the prior design and establishes a basis for conducting a sequence of subsequent examinations.

### 3.2. Power Capacity Testing

Next, power capacity experiments are conducted for the assembled RA prototypes. The experimental block diagram is shown in Figure 8. An S-band relativistic magnetron, Tektronix TDS7404 oscilloscope, power meter, and other instruments are adopted in this experiment. The experiment process is as follows: two open-ended waveguides, A and B, are positioned on either side of the maximum radiation direction CD of the feed horn. The open-ended waveguide B is fixed to monitor the stability of the HPM source (S-band relativistic magnetron). The open-ended waveguide A measures the power density at various positions by changing its location. The furthest distance between the phase center of the feed horn and the open-ended waveguides is approximately 5 m. Then, two different types of RAs are placed in sequence at the calibrated position of the open-ended waveguide A, and the utilization of an infrared camera facilitates the observation of surface flashover on the RA surface, enabling the determination of any potential breakdown in the RA surface.

The power density *P_d_* of the open-ended waveguide A at the position of #*i* (*i* = 0, 1, …) is calculated by the following equation:(2)$$P_{d}=\frac{P_{i}}{S_{eff}}$$
where *S_eff_* is the effective receiving area of the open-ended waveguide A (the BJ32 rectangle waveguide), and *P_i_* is the microwave power received by the open-ended waveguide A at the position of #*i*, which is calculated by
(3)$$P_{i}=10^{\left(10\cdot \mathrm{lg}\frac{V^{2}}{2R}+A\right)/10}$$
where *V* is the microwave amplitude received by the open waveguide A, *A* is the total attenuation including transmission cables and attenuators, and *R* is the distance between the transmitting and receiving antennas.

According to Equations (2) and (3), the power density of each position at 2.8 GHz in Figure 8 is shown in Table 2. It can be seen that from the position of #0 to the position of #12, the power density decreases, and #13 to #16 are broadside positions, and the power density increases. The change in detection voltage of the open-ended waveguide B is minimal, indicating the stable state of the HPM source. Furthermore, the microwave signals collected by the open-ended waveguides can be displayed on the Tektronix TDS7404 oscilloscope, as shown in Figure 9. It can be observed that the microwave pulse widths of the received signals are approximately 40 ns, and the resonance frequency is 2.8 GHz.

The centers of the two RA surfaces are gradually moved from the low-power density position to the high-power density position. For the RA with element I, the camera images of the RA surface at the position of #12 in Table 2 are shown in Figure 10. Figure 10a,b depict the active and inactive states of the HPM, respectively. By comparing the two figures, it can be observed that there are no abnormal phenomena on the RA surface. This suggests that the RA with element I does not reach the breakdown threshold at this lower power position.

Similar results are obtained at nearby locations until the RA with element I is placed at the position of #5, as shown in Figure 11. When the HPM is activated, sporadic flashover phenomena can be observed on the right side of the RA surface, which indicates that a few elements of the RA surface have been broken down at this position. When the RA surface is placed at the position of #3 and #1, there are variations in the RA surface captured by the camera, also depicted in Figure 12 and Figure 13, respectively. The results indicate that the surface flashover area is gradually expanding. Especially for the position of #1, the entire RA surface undergoes a flashover phenomenon, which means that more elements on the RA surface are broken down under HPMs with the increase in microwave power density.

According to the same measured method, the power capacity test can be conducted on the RA surface with element II. The images of the RA surface with element II at the position of #16 are captured by the camera and are shown in Figure 14. It can be seen that there is no flashover on the RA surface when the HPM is active, which indicates that the RA with element II has not yet reached its breakdown threshold. However, due to the limited power of the HPM source, a higher power test has not been carried out on the RA with element II.

### 3.3. Maximum Directivity Testing under High Power

In the previous section, it is observed that no flashover phenomenon occurs on the surface of the RA with element II. However, it is inconclusive whether the structural integrity of the element remains unaffected when subjected to a power level of 1 GW. Therefore, in this section, a antenna directivity test is conducted under HPM conditions. The block diagram of the test is shown in Figure 15, where the microwave injected into the feed horn comes from the HPM source. In the event that the RA surface is positioned at #16 and the HPM is activated, any internal damage incurred within the RA surface would lead to the impairment of its phase modulation capability. In addition, due to the metal ground, the antenna radiation beam would be reflected in the mirror direction of the incident wave. Otherwise, it persists in propagating along the normal direction of the RA surface. The setup for the measurement is shown in Figure 16. Upon conducting the experiment, it has been determined that the power density in the normal and mirror directions of the RA are 221,238 W/cm^2^ and 14,600 W/cm^2^, respectively. These results imply that the RA with element II can maintain its ability to radiate beams in the expected direction at the position of #16 (the power capacity is about 1 GW).

## 4. Conclusions

This paper proposes two RA elements and conducts detailed high-power experimental research for RAs using relativistic magnetrons, accelerators, and other equipment. By observing whether flashover occurs on the RA surface, the power capacity threshold of the RAs can be effectively determined. The power capacity thresholds of two types of antennas are evaluated using this method. It is worth noting that the RA based on element II enables it to effectively maintain the desired radiation direction, even when operating at a power level of 1 GW. This advancement significantly enhances the potential utilization of RAs within HPM systems.

## Figures and Tables

**Figure 1 micromachines-15-00399-f001:**
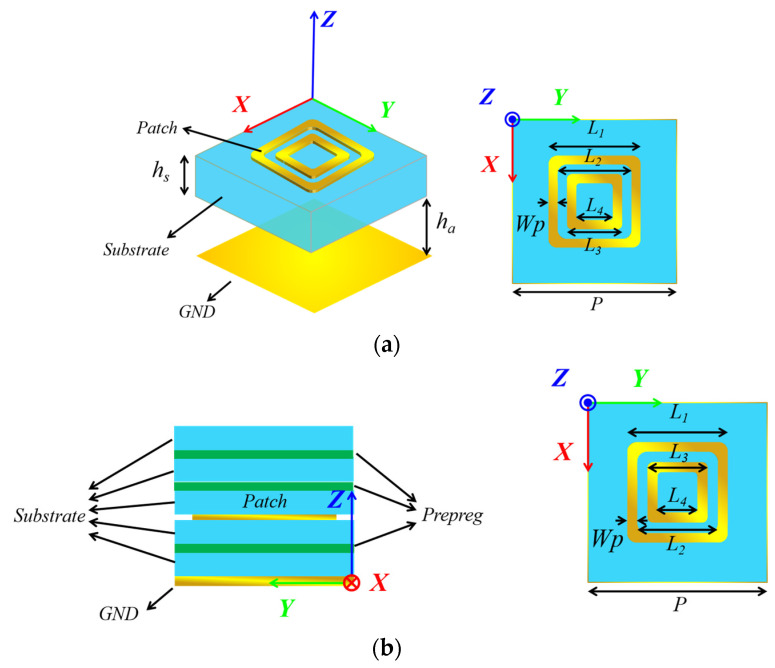
Structure of the two RA elements. (**a**) RA element I; (**b**) RA element II.

**Figure 2 micromachines-15-00399-f002:**
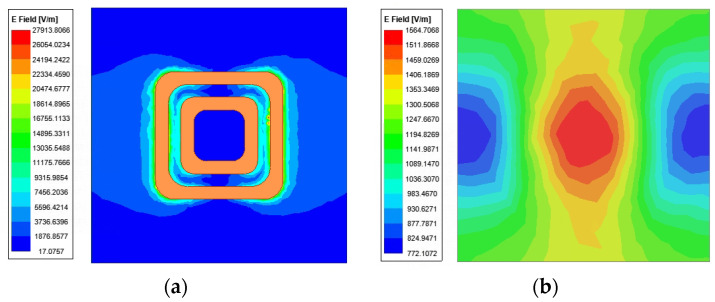
Surface electric field distribution of (**a**) RA element I; (**b**) RA element II.

**Figure 3 micromachines-15-00399-f003:**
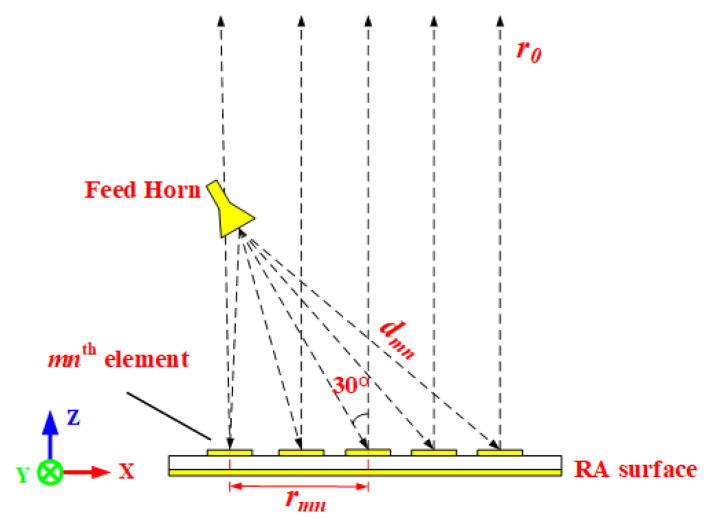
The configuration of the RA for HPM.

**Figure 4 micromachines-15-00399-f004:**
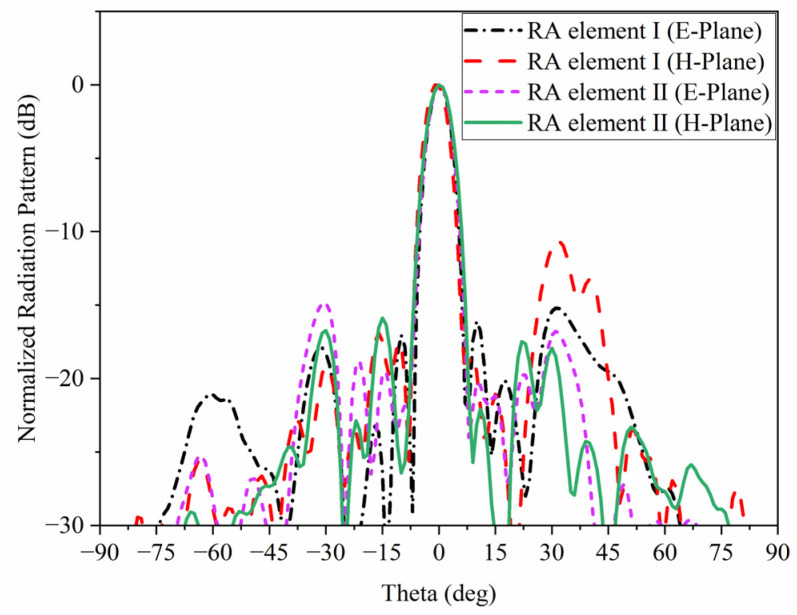
Normalized radiation patterns of RA with two types of elements.

**Figure 5 micromachines-15-00399-f005:**
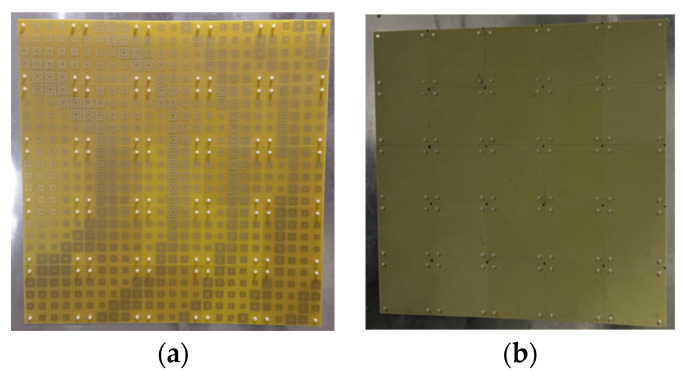
Prototype of the RA surface with (**a**) RA element I and (**b**) RA element II.

**Figure 6 micromachines-15-00399-f006:**
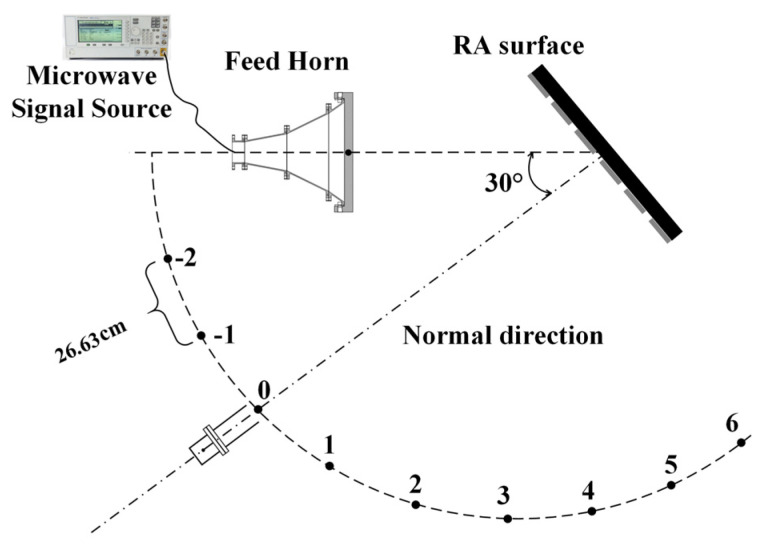
The testing block diagram of the patterns of two RAs.

**Figure 7 micromachines-15-00399-f007:**
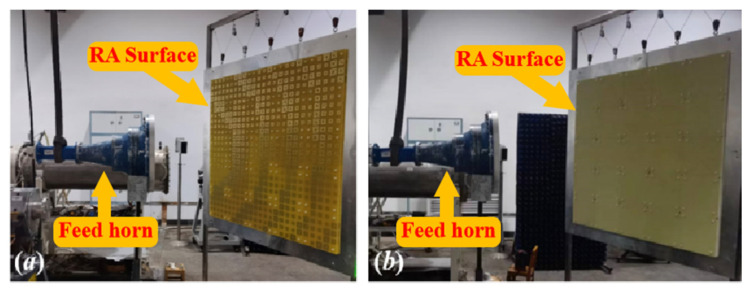
The experimental setups of the RA with (**a**) RA element I and (**b**) RA element II.

**Figure 8 micromachines-15-00399-f008:**
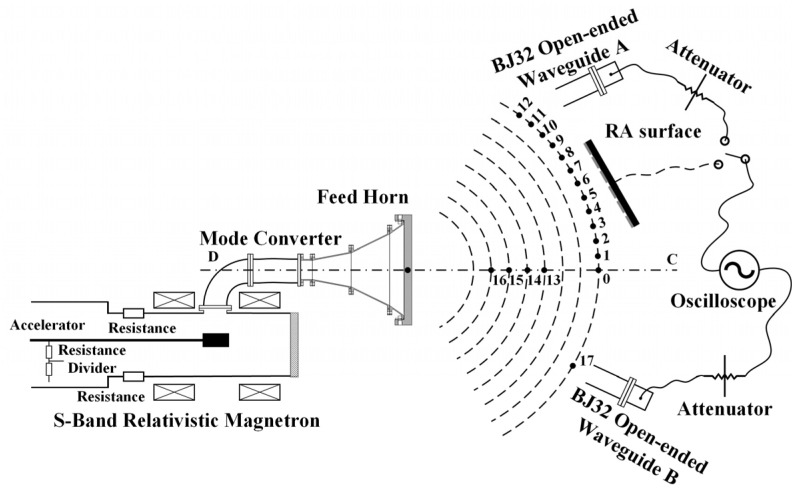
The testing block diagram of the power capacity of the RAs.

**Figure 9 micromachines-15-00399-f009:**
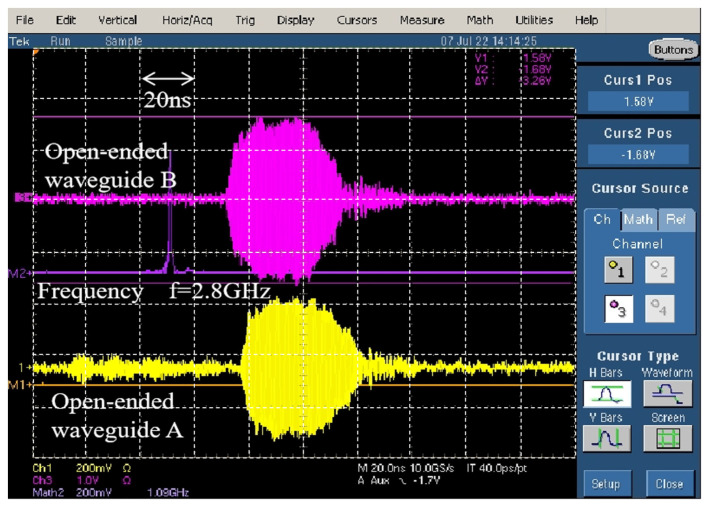
Waveform of microwave pulse and its spectrum.

**Figure 10 micromachines-15-00399-f010:**
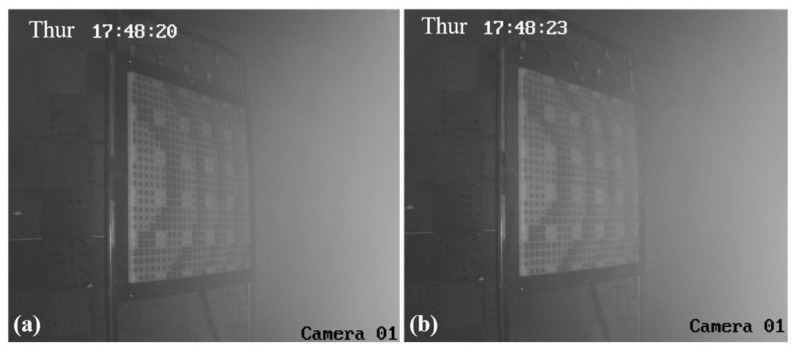
The camera images of RA surface with element I at the position of #12. (**a**) HPM is inactive; (**b**) HPM is active.

**Figure 11 micromachines-15-00399-f011:**
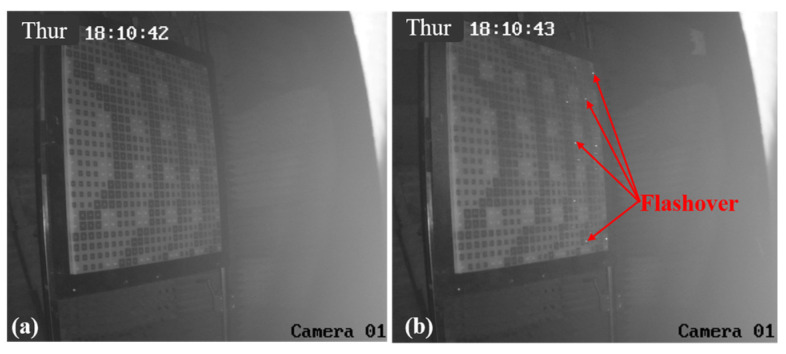
The camera images of RA surface with element I at the position of #5. (**a**) HPM is inactive; (**b**) HPM is active.

**Figure 12 micromachines-15-00399-f012:**
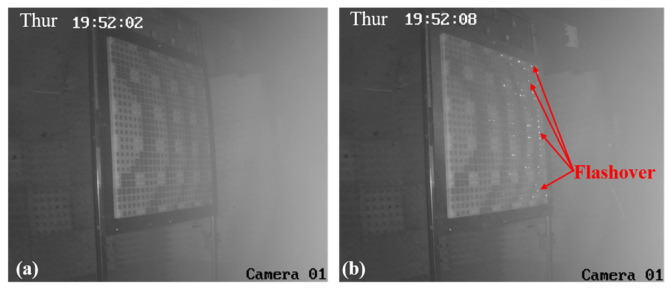
The camera images of RA surface with element I at the position of #3. (**a**) HPM is inactive; (**b**) HPM is active.

**Figure 13 micromachines-15-00399-f013:**
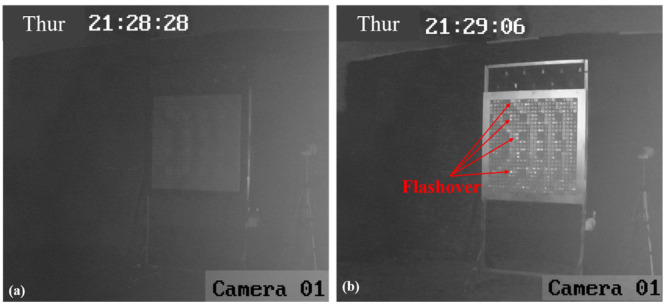
The camera images of RA surface with element I at the position of #1. (**a**) HPM is inactive; (**b**) HPM is active.

**Figure 14 micromachines-15-00399-f014:**
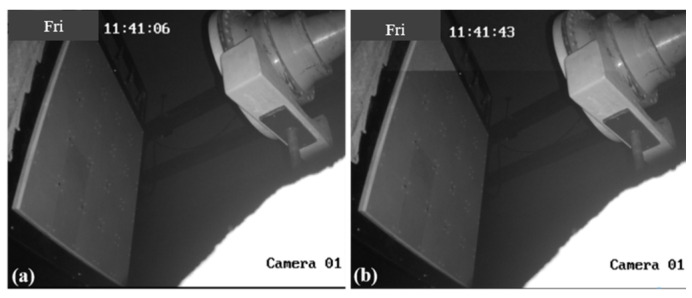
The camera images of RA surface with element II at the position of #16. (**a**) HPM is inactive; (**b**) HPM is active.

**Figure 15 micromachines-15-00399-f015:**
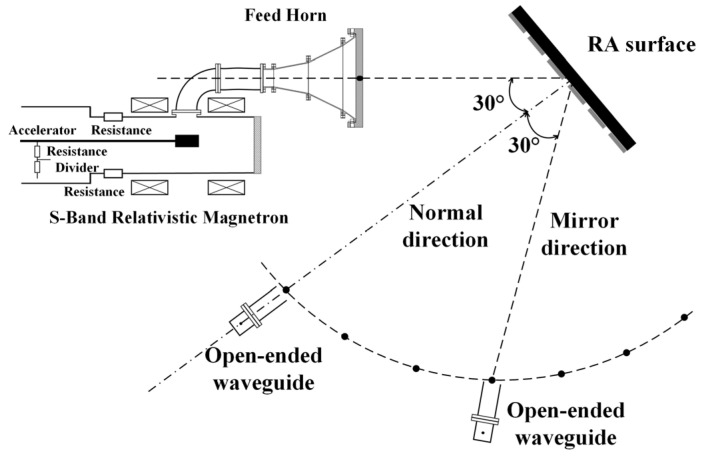
Block diagram of directivity testing of the RA with element II under HPM.

**Figure 16 micromachines-15-00399-f016:**
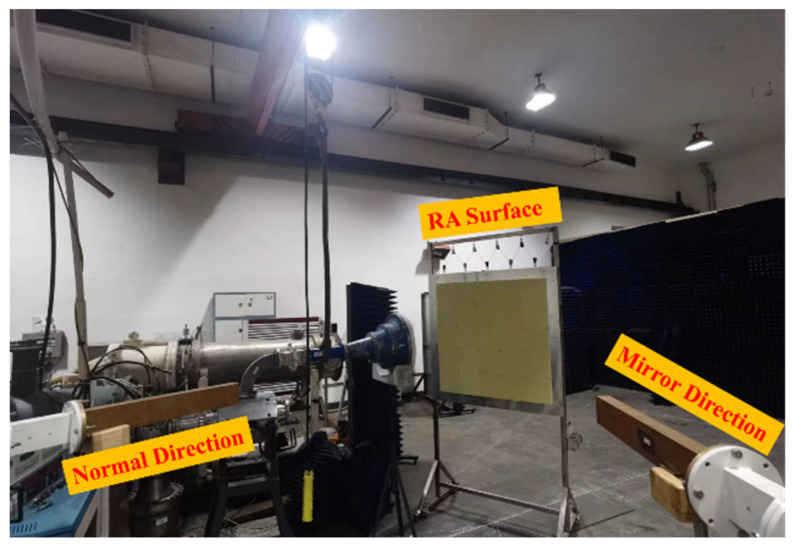
Measured setup of the directivity testing under HPM.

**Table 1 micromachines-15-00399-t001:** The results of directivity testing.

Testing Position	Angle Information (deg)	RA with Element I (dBm)	RA with Element II (dBm)
#−1	10	−35.46	−34.15
#0	0	−23.3	−22.1
#1	10	−29.7	−28.53
#2	20	−30.3	−29.02
#3	30	−34.7	−28.9
#4	40	−36.8	−31.24
#5	50	−44.2	−33.64
#6	60	−54.1	−42.3

**Table 2 micromachines-15-00399-t002:** The results of power density calibration.

Reference Position	Angle Information (deg)	Distance between Each Position and Position #0 (m)	The Voltage of the Open-Ended Waveguide B(V)	The Voltage of the Open-Ended Waveguide A(V)	The Power Density of the Open-Ended Waveguide A *P_d_* (W/cm^2^)
#0	0	0	0.612	2.91	24,412
#1	5	0.44	0.6332	2.66	21,067
#2	10	0.87	0.628	2.63	19,800
#3	15	1.31	0.64	2.11	13,440
#4	20	1.74	0.52	1.63	7748
#5	25	2.16	0.53	1.34	5236
#6	30	2.59	0.61	1.26	4556
#7	35	3.01	0.57	1.08	3473
#8	40	3.42	0.57	0.93	2575
#9	45	3.83	0.51	0.64	730
#10	50	4.23	0.54	0.53	810
#11	55	4.62	0.61	0.426	511
#12	60	5	0.61	0.272	210
#13	-	1.8	0.62	4.345	54,200
#14	-	2.4	0.61	5.146	77,200
#15	-	3	0.58	5.401	85,000
#16	-	3.6	0.592	6.044	106,505

## Data Availability

Data are available in a publicly accessible repository.

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
