# Peer review of "Experimental Investigation of Reflectarray Antennas for High-Power Microwave Applications"

_micromachines, 2024, doi:10.3390/mi15030399_

Round 1
Reviewer 1 Report
Comments and Suggestions for Authors
Manuscript is in good form. However, following comments may further improve the quality of paper for readers.
1. Provide reference for Equation 1.
2. Authors wrote, "the breakdown threshold in air is 3 MV/m" (Line number 124). Provide reference to this statement.
3. How are the theoretical power capacity and power density of the element calculated? Please provide a systematic elaboration of the procedure using Equation 1.
Reviewer 2 Report
Comments and Suggestions for Authors
The focus on power capability is interesting but the work provides little to no theoretical background and the thermal pictures are very difficult to make anything from. Therefore, as an overall assessment the work fails to clearly present novel contribution(s). The lack of theory means that the new RA design is "pulled from a hat". As an engineer, if I was given a specific design task, to implement an RA solution to meet a specific power handling level, then the presented work would be only of very limited value. The many measurements are greatly appreciated but - as already mentioned - the thermal pictures do not really provide value.

In general the work is well presented. There are some minor issues, some of which (most?) I have indicated in the attached file. Here you may also see my general remarks throughout the paper.
Reviewer 3 Report
Comments and Suggestions for Authors
The author proposed a reflectarray antenna (RA) for high power microwave applications. Numerical and experimental investigation (based on the S-band relativistic magnetron, 133 Tektronix TDS7404 oscilloscope) of power capacity and radiation directivity are conducted for two RA protypes in different angular positions. They further demonstrate the stability and the practicality of the proposed devices by the high-power pulse reproduction and high-power directivity maintenance study. The following revision should be made before this paper can be accepted as publishable.
1. For the numerical study, the author only provides the near-field distribution upon resonance which demonstrates the power capacity. However, for the other aspect, a far-field radiation pattern might be necessary to complete the investigation.
2. In section 4.1, sentence “Based on the configuration in FigureFigure 3, it can be observed that the electromagnetic waves emitted by 120 the feed source will propagate outward at a perpendicular angle to the RA surface” has two “figure3” and need to be removed.
3. The author might export the data in Figure 8 and replot it for clarification.
4. For the directivity test under high power, author might provide some far-field data or plot as a correspondence to the numerical study (radiation pattern, angular-dependent radiation power, etc.)
Comments on the Quality of English LanguageFluent enough and express the idea clearly
Round 2
Reviewer 2 Report
Comments and Suggestions for Authors
Dear sirs
Thank you for the revised version and your response/comments. I have tried to adresse everything here in combination with the attached file.
Response: You state that the contribution lies in the measurement and that such measurements are rarely seen. I agree that it is rare to see such measurements but the paper does not indicate that this is the focus. Looking at the abstract there is a clear focus on the RAs. The same for the paper content. If indeed you feel that the measurement part is the core then the paper should be restructured significantly. Also, more attention to measurement limitations and the quality of the pictures is needed then. In my view the paper is weakest in the validation part in its current form.
Comment 1: Noted and appreciated.
Comment 2: You miss my point. What SI unit is used to quantify terms like "good", "superior" and "excellent"? I expect more accuracy in terminology from a scientific paper.
Comment 6: I see your point but the thing is that in the paper you refer to phase linearity using figures where L1 is the x-axis parameter. The linearity in this regard is completely different from the actual operational performance linearity that you describe in you comment. Linearity across frequency (and potentially also incident power) is important .. and this is not the linearity that you indicate in the figures. In fact, the phase response change for your two RA elements is larger for your revised design when comparing L1 = 33 for the conventional and L1 = 17 for the revised design. Also, returning to the core of the work .. if the measurement part is the contribution then it is completely irrelevant to spend so much effort on discussing anything but power handling.
Comment 7: I realise that .. but looking only at the phase for a RA makes no sense as the combined complex reflection performance is what defines the overall performance in the end. From a design point you need to have the ability to have elements that can provide between 0 - 360 degrees in phase shift but the linearity of the (x,y) = (L1, phase shift) curve is irrelevant whereas the magnitude of the reflection as function of incident angle is relevant.
Comment 8: Having such details in the paper makes for a much better read as no information is "hidden" from the reader.
Comment 9: The reasoning here still does not add up. I have put comments in the file.
Comment 11: Of course .. but perhaps make sure that the distinction is clear.
Comment 12: Please add such details to the figures for completeness.
Comment 15: I stand by my comment from the first version as the figures are really difficult to see/read anything from. And again .. if this is to be the core of the contribution then it ought to be significantly improved.

Reviewer 3 Report
Comments and Suggestions for Authors
After the revision, the manuscript looks much better. For the far-field data that the author cannot disclose, a proper reason has been provided.
Round 3
Reviewer 2 Report
Comments and Suggestions for Authors
Dear sirs
Thank you for your revised paper and your comments. I do apologize for the hand written comments but this makes reviewing much less time consuming. With the typed remarks I get the impression that it has not been all too bad for you though.
I have added comment to the files and here as well. The typed remarks here should cover all notes in the files also.
A - If this is where you think the novelty is then then paper really needs to focus on this. It is better with the last version but I still think that the design and optimization of the RA's should be significantly reduced, to leave room for much more focus on the measurement side of the work. Then maybe a second paper on the design would be in place.
B - But the methodology here is not convincing. Focus on one issue and do this well. Don't focus on two issues and then end up with something less good.
C - The fact that other people use non-scientific terminology can never be an argument for doing the same.
D - Your figure (Fig. 12a)) suggest that most elements are large, hence around L1 = 33 mm. Also, the location of the different-sized elements on the overall array has an impact on the resulting performance.
E - Phase and magnitude response matters. One may be more significant that the other but this does not imply that one can just be disregarded at will. You need to provide suitable argumentation and not simply refer to what others do or do not do.
F - I was not implying that you were intentionally hiding information. But by not providing the information then, to the reader, the information is hidden. If you think that I was suggesting that you hide information on purpose then I apologize. This was not my intention nor my point.
G - Comparing the reflection magnitudes then the revised RA is generally lower (i.e. worse) than the ordinary RA. This would make sense seeing that absorption losses are added to the revised RA. The paper provides no reasonable or convincing explanation for why the resulting array performance of the revised design is better.
H - Much easier to see with the much larger pictures and also with the addition of arrows.
I - This difference is already included in the results given in Fig. 3 and Fig. 8. Fig. 3 shows a reflection magnitude of around -0.25 to -1.25 dB on average. Fig. 8 shows around -0.75 to -1.5 dB on average. Therefore, on average the loss of the revised RA is higher. Fig. 13 shows only a difference of 0.75 dB, so not all too significant. However, Fig. 3 and Fig. 8 do not support your argument so the difference must be for some other reason. What about measurement inaccuracy?
